# Value-driven Hindsight Modelling

**Arthur Guez**      **Fabio Viola**      **Théophane Weber**      **Lars Buesing**

**Steven Kapturowski**      **Doina Precup**      **David Silver**      **Nicolas Heess**

DeepMind
aguez@google.com

## Abstract

Value estimation is a critical component of the reinforcement learning (RL) paradigm. The question of how to effectively learn value predictors from data is one of the major problems studied by the RL community, and different approaches exploit structure in the problem domain in different ways. Model learning can make use of the rich transition structure present in sequences of observations, but this approach is usually not sensitive to the reward function. In contrast, model-free methods directly leverage the quantity of interest from the future, but receive a potentially weak scalar signal (an estimate of the return). We develop an approach for representation learning in RL that sits in between these two extremes: we propose to learn what to model in a way that can directly help value prediction. To this end, we determine which features of the *future* trajectory provide useful information to predict the associated return. This provides tractable prediction targets that are directly relevant for a task, and can thus accelerate learning the value function. The idea can be understood as reasoning, in hindsight, about which aspects of the *future* observations could help *past* value prediction. We show how this can help dramatically even in simple policy evaluation settings. We then test our approach at scale in challenging domains, including on 57 Atari 2600 games.

## 1 Introduction

Consider a baseball player trying to perfect their pitch. The player performs an arm motion and releases the ball towards the batter, but suppose that instead of observing where the ball lands and the reaction of the batter, the player only hears the result of the play in terms of points or, worse, only the final result of the game. Improving their pitch from this experience appears hard and inefficient, yet this is essentially the paradigm we employ when optimizing policies in model-free reinforcement learning. The scalar feedback that estimates the return from a state (and action), encoding *how well things went*, drives the learning while the accompanying observations that may explain that result (e.g., flight path of the ball or the way the batter anticipated and struck the incoming baseball) are ignored. To intuitively understand how such information could help value prediction, consider a simple discrete Markov chain $X \to Y \to Z$, where $Z$ is the scalar return and $X$ is the observation from which we are trying to predict $Z$. If the space of possible values of $Y$ is smaller than $X$, then it may be more efficient to estimate $P(Y|X)$ and $P(Z|Y)$ rather than directly estimating $P(Z|X)$.[1] In other words, observing and then predicting $Y$ can be advantageous compared to directly estimating the signal of interest $Z$. Model-based RL approaches would duly exploit the observed $Y$ (by modelling $P(Y|X)$), but $Y$ would, in general scenarios, contain information that is irrelevant to $Z$ and hard to predict. Building a full high-dimensional predictive model to indiscriminately estimate all possible future observations, including potentially chaotic details of the ball trajectory and the spectators' response, is a challenge that may not pay off if the task-relevant predictions (e.g., was the throw accepted, was

the batter surprised) are error-ridden. Model-free RL methods directly focus only on the relation of $X$ to $Z$, rather than attempting to learn the full dynamics. These methods have recently dominated the literature, and have attained the best performance in a wide array of complex problems with high-dimensional observations [12, 17, 8, 7].

In this paper, we propose to augment model-free methods with a lightweight model of future quantities of interest. The motivation is to model only those parts of the future observations ($Y$) that are needed to obtain better value predictions. The major research challenge is to learn, from observational data, which aspects of the future are important to model (i.e. what $Y$ should be). To this end, we propose to learn a special value function *in hindsight* that receives future observations as an additional input. This learning process reveals features of the future observations that would be most useful for value prediction (e.g., flight path of the ball or reaction of the batter), if provided by an oracle. We then learn a model that predicts these hindsight features using only information available at test time (at the time of releasing the ball, we knew the identity of the batter, the type of throw and spin of the ball). Learning these value-relevant features can help representation learning for an agent and provide an additional useful input to its value and policy functions. Experimentally, we show that hindsight value functions surpassed model-free RL methods in a challenging association task (Portal Choice). When added to the prior state-of-the-art model-free RL method for Atari games [11], hindsight value functions significantly increased median performance, from 833% to 965%.

## 2   Background and Notation

Consider a reinforcement learning (RL) agent interacting in a sequential decision-making environment [19]. At each step $t$, after observing state $s_t$,[2] the agent outputs an action $a_t \sim \pi(A|s_t)$, and obtains a scalar reward $R_t$ and the next-state $s_{t+1}$ from the environment. The sum of discounted rewards from $s$ is the return denoted by $G = \sum_{t=0}^{\infty} \gamma^t R_t$, with $\gamma \in (0,1)$. Its expectation is called the value function, $v^\pi(s) = \mathbb{E}_\pi[G|S_0 = s]$. An important related quantity is the action-value, or Q-value, which corresponds to the same expectation with a particular action executed first: $q^\pi(s,a) = \mathbb{E}_\pi[G|S_0 = s, A_0 = a]$. The agent's goal is to adapt the policy $\pi$ in order to achieve a higher value $v^\pi$. This usually entails learning an estimate of $v^\pi$ for the current policy $\pi$, the problem we focus on in this paper. From now on, we drop $\pi$ from the notation for simplicity.

**Direct (Model-free) Learning**    A common approach to estimate $v$ (or $q$) is to represent it as a parametric function $v_\theta$ (or $q_\theta$) and directly update its parameters based on sample returns of the policy of interest. Value-based RL algorithms vary in how they construct a value target $U$ from a single trajectory. They may regress $v_\theta$ towards the Monte-Carlo (MC) return ($U_t = G_t$), or exploit sequentiality by relying on a form of temporal-difference learning to reduce variance (e.g., the TD(0) target $U_t = R_t + \gamma v_\theta(S_{t+1})$). For a given target definition $U$, the value loss $\mathcal{L}_v$ to derive an update for $\theta$ is: $\mathcal{L}_v(\theta) = \frac{1}{2}\mathbb{E}_s[(v_\theta(s) - U)^2]$. In constructing a target $U_t$ based on a trajectory of observations and rewards from time $t$, the observations are either unused (for a MC return) or only indirectly exploited (when bootstrapping to obtain $U$, see Sec. 3.4). In all cases, the trajectory is distilled into a scalar signal that estimates the return, and other relevant aspects of future observations are discarded. In domains with high-dimensional observation spaces or partial observability, it can be particularly difficult to discover correlations with this noisy, possibly sparse, signal.

**Model-based Learning**    An indirect way to estimate values is to first learn a model of the dynamics. For example a 1-step observation model $m_\theta$ learns to predict the conditional distribution $s_{t+1}, r_t|s_t, a_t$. Then a value estimate $v(s)$ for state $s$ can be obtained by autoregressively rolling out the model (until the end of the episode or to a fixed depth with a parametric value bootstrap). The model is trained on potentially much richer data than the return signal, since it uses all information in the trajectory. Indeed, the observed transitions between states can reveal the structure behind a sparse reward signal. A drawback of classic model-based approaches is that they predict a high-dimensional signal, a task which may be costly and harder than directly predicting values. As a result, the approximation of the dynamics $m_\theta$ may contain errors where it matters most for predicting values [21]. Although the observations carry all the data from the environment, most of it is not essential for the task [6]. The concern that modelling all observations is expensive also applies when the model is not used for actual rollouts but merely for representation learning.

# 3 Hindsight Value Functions

While classic model-based methods fully use the high-dimensional observations at some cost, model-free methods focus only on the most relevant low-dimensional signal (the scalar return). We propose a method that strikes a balance between these paradigms.

## 3.1 Hindsight value and model

We first introduce a new quantity, the *hindsight* value function $v^+$, which depends on quantities only available at training time. This value still represents the expected return from a state $s_t$, but it is conditioned on additional information $\tau_t^+ \in \mathbb{T}^+$ occuring after time $t$: $v^+(s_t, \tau_t^+) = \mathbb{E}[G|S_0 = s_t, \mathrm{T}^+ = \tau_t^+]$; $\tau_t^+$ can be defined to include any of the future observations, actions and rewards occurring in the trajectory following $s_t$. The hindsight value can be seen as an instance of a general value function in a stochastic computational graph as defined by Weber et al. [25].

In this paper, we focus on the case of $\tau^+$ containing $k$ additional observations $\tau_t^+ = s_{t+1}, s_{t+2}, \ldots s_{t+k}$ occurring after time $t$: $v^+(s_t, \tau_t^+) = \mathbb{E}[G|S_0 = s_t, \ldots, S_k = s_{t+k}]$. Furthermore, we require estimates of $v^+$ to follow the following parametric structure: $v^+(s_t, \tau_t^+; \theta) = \psi_{\theta_1}(f(s_t), \phi_{\theta_2}(\tau_t^+))$, where $\theta = (\theta_1, \theta_2)$, which forces information about the future trajectory through some vector-valued function $\phi \in \mathcal{R}^d$. Intuitively, $v^+$ is estimating the expected return from a past time point using privileged access to future observations. Note that if $k$ is large enough, then $v^+$ may simply estimate the empirical return from time $t$ given access to the state trajectory. However, as we will show, if $k$ is small and $\phi$ is low-dimensional, $\phi$ can become a bottleneck representation of the future trajectory $\tau_t^+$. Hence, by learning in hindsight, we identify features that are maximally useful to predict the return from time $t$.

The hindsight value function is not a useful quantity by itself, since we cannot readily use it at test time, because of its use of privileged future observations. It cannot be used as a baseline for policy gradient either, as it will yield a biased gradient estimator [25]. Instead, we propose to learn a model $\hat{\phi}$ of $\phi$, that can be used at test time. We conjecture that if privileged features $\phi$ are useful for estimating the value, then the model of those features will also be useful (and at least a good signal for representation learning). We propose to learn the approximate expectation model $\hat{\phi}_{\eta_2}(s)$ conditioned on the current state $s$ and parametrized by $\eta_2$, by minimizing an appropriate loss $\mathcal{L}_{\mathrm{model}}(\eta_2)$ between $\phi$ and $\hat{\phi}$ (e.g., a squared loss). The approximate model $\hat{\phi}$ can then be leveraged to obtain a better model-based value estimate $v^m(s; \eta) = \psi_{\eta_1}(f(s), \hat{\phi}_{\eta_2}(s))$. Although $\hat{\phi}(s)$ cannot contain more information than included already in the state $s$, it can still benefit from training with a richer signal (i.e., privileged value-relevant features $\phi$) before the value converges.

This formulation can be straightforwardly extended to the case where observations are provided as inputs, instead of full states: a learned function then outputs a state representation $h_t$ based on the current observation $o_t$ and previous estimated state $h_{t-1}$ (details in Sec. 4). Fig. 2 summarizes the relation between the different quantities in this *hindsight modelling* (HiMo) approach.

## 3.2 Illustrative example

To understand how the approaches of estimating the value function differ and how value-driven hindsight modelling might be beneficial, consider the Markov Reward Process in Fig. 1. Each episode consists of a single transition from initial state $s$ to terminal state $s'$, with a reward $r(s, s')$ on the way. The key aspect of this domain is that observing $s'$ reveals structure that helps predict the value at $s$. Namely, part of $s'$ provides a simplified view of some of the information present in $s$, which is directly involved in predicting $r$ (in the baseball narrative, this could correspond to observing the final configuration between the ball and bat as they collide). Full details of the domain can be found in Appendix A.3.

Let us consider how the different value learning approaches presented above fare in this problem. For direct learning, the value from $v(s')$ is 0 since $s'$ is terminal, so any n-step return is identical to the MC return, that is, the information present in $s'$ is not leveraged. Results of learning $v$ from $s$ given the return are presented in Fig. 1 (model-free, orange curve). A model-based approach first predicts $s'$ from $s$, then attempts to predict the value given $s$ and the estimated next state. When increasing the input dimension, given a fixed capacity, the model does not focus its attention on the reward-relevant

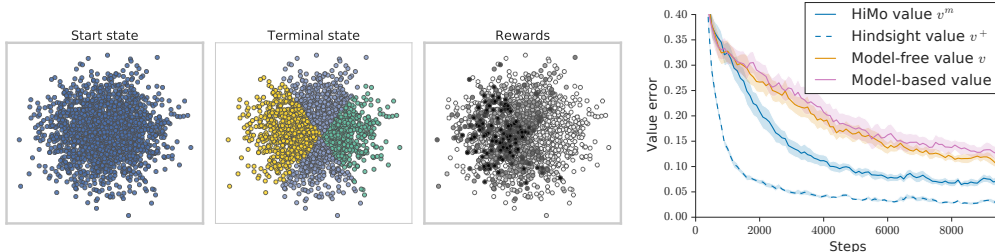

Figure 1: **Illustrative example of Sec. 3.2 (Left)** Visualization of trajectories. Model-free value prediction see the start state $s$ on the left (which are normally distributed) and must predict the corresponding color-coded reward on the right (larger is darker). Hindsight value prediction can leverage the observed structure in the terminal state $s'$ (middle) to obtain a better value prediction; $s'$ is plotted superimposed on $s$ color-coded with the discrete reward-relevant quantity in $s'$. States have dimension $D=4$ and are projected in 2D in this example. **(Right)** Learning the value of the initial state for various methods. The data dimension is $D = 32$ for this experiment, and the useful data dimension in $s'$ is 4. The results are averaged over 4 different instances, each repeated twice. Note that $v^+$ (dotted line) is using privileged information (the next state), so it is a form of performance bound.

structure in $s'$ and makes errors where it matters most. As a result, it can struggle to learn $v$ faster than a model-free estimate (pink curve in Fig. 1). When learning in hindsight, $v^+$ can directly exploit the revealed structure in the observation of $\tau^+$ (this corresponds to terminal state $s'$ in this case), and as a result, the hindsight value (dashed blue curve) learns faster than the regular causal model-free estimate. This drives the learning of $\phi$ and its model $\hat{\phi}$, which directly gets trained to predict these useful features for the value. As a result, $v^m$ also benefits and learns faster than the regular $v$ estimate on this problem (blue curve).

## 3.3  When is it advantageous to model in hindsight?

To understand the circumstances in which hindsight modelling provides a better value estimate, we provide the following result:

**Proposition.** *Suppose that $v_\theta^m$ is sharing the same function $\psi$ as $v^+$ (i.e., $\theta_1 = \eta_1$), and let $\psi$ be linear ($\psi_{\theta_1}(f, \phi) = \left( \begin{smallmatrix} \omega_1 \\ \omega_2 \end{smallmatrix} \right)^\top \left( \begin{smallmatrix} f \\ \phi \end{smallmatrix} \right) + b$, where $\theta_1 = (\omega_1, \omega_2)$). Assume a squared loss for the model and the following relation between value losses: $\mathcal{L}(v^+) = C\mathcal{L}(v)$ with $C < 0.5$ (i.e., estimating the value with more information is an easier learning problem). Then, the following holds: $\mathcal{L}_{model}(\eta_2) < \frac{(1-2C)\mathcal{L}(v)}{2\|\omega_2\|^2} \implies \mathcal{L}(v^m) < \mathcal{L}(v)$. (Proof in appendix.)*

Intuitively, this relates how small the modelling error needs to be in order to guarantee that the value error for $v^m$ is smaller than the value error for the direct estimate $v$. The modeling error can be large for different reasons. If the environment or the policy is stochastic, then there is some irreducible modelling error for the deterministic model. Even in such cases, a small $C$ can make hindsight modelling advantageous. The modeling error could also be high because predicting $\phi$ is hard. For example, it could be that $\phi$ essentially encodes the empirical return, which means predicting $\phi$ is at least as hard as predicting the value function ($\mathcal{L}_{model}(\eta_2) \geq \mathcal{L}(v)$). Or, it could be that $\phi$ is high-dimensional, causing both a hard prediction problem and a decrease in the acceptable threshold for $\mathcal{L}_{model}$ (since $\|\omega_2\|^2$ will grow). We address some of these concerns with specific architectural choices, such as allowing $v^+$ a limited view on future observations and having low dimensional $\phi$ (see Sec. 4). The analysis above ignores any advantage obtained from representation learning when training $\hat{\phi}$ (for shared state encoding functions).

## 3.4  Relation to bootstrapping

It is worth emphasizing the difference between hindsight modelling and the bootstrapping typically done in temporal difference learning. Both exploit a partial trajectory for value prediction, but have different objectives and mechanisms. Bootstrapping helps to provide potentially better value targets from a trajectory (e.g., to reduce variance or deal with off-policyness), but it does not give a richer training signal in itself as it does not communicate more information about the future than a return

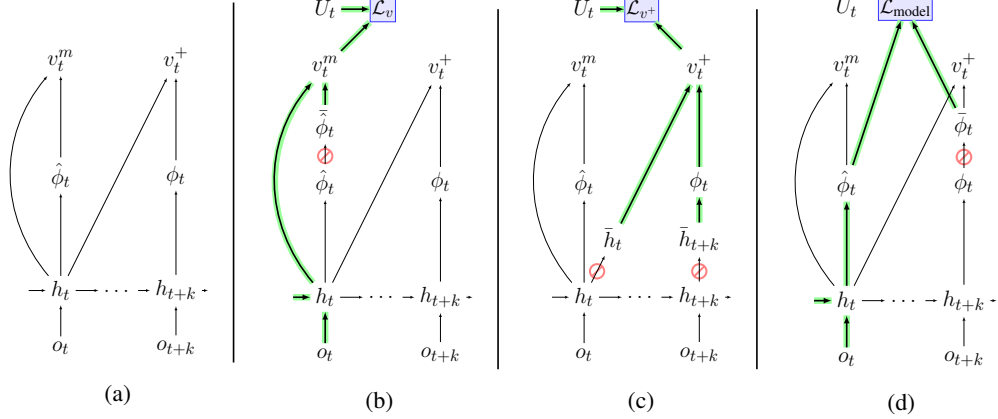

Figure 2: **HiMo architecture.** (a) depicts the overall HiMo architecture, where $v_t^m$ is the model-augmented value prediction available at test time $t$ ($\hat{\phi}_t$ is a model of $\phi_t$), and $v_t^+$ is the hindsight value for step $t$ that uses additional privileged observations (up to $o_{t+k}$). Nodes are tensors and edges are (learnable) tensor transformations, e.g. neural networks. The panels (b), (c) and (d) illustrate the different learning losses, where green identifies learned edges, and the red symbol Ø denotes that gradients are stopped in backpropagation. In particular, (b) shows the subset of the network used for the value function loss $\mathcal{L}_v$, (c) the hindsight value function loss $\mathcal{L}_{v^+}$ and (d) the model loss $\mathcal{L}_{\mathrm{model}}$. $U_t$ denotes the value target at time $t$.

statistic (a scalar value). Consider for example a *deterministic* scenario where we want to estimate the value of a state $s_0$ drawn from some distribution, and $v_\theta(s_t) = v(s_t), \forall t > 0$, i.e., all subsequent value estimates are perfect. Hence, the MC return and all the $n$-step returns from any $s_0$ are equal, so bootstrapping has no consequence. Yet, there might still be some useful information in the future trajectory which would accelerate the learning of $v_\theta(s_0)$, if predicted at $s_0$. For example, suppose $s_2$ has some bits correlated with the trajectory's final return, which are easy to predict from $s_0$. Hindsight modeling has precisely the potential of learning such a vector-valued feature of $s_2$. One concrete example can be found in Appendix section A.1.

## 4   Architecture

We now describe an architecture for HiMo that we found to work at scale and that we tested in Sec. 6. To deal with partial observability, we employ a recurrent neural network, the state-RNN, which replaces the state $s_t$ with a learned internal state $h_t$, a function of the current observation $o_t$ and past observations through $h_{t-1}$: $h_t = f(o_t, h_{t-1}; \eta_3)$, where we have extended the parameter description of $v^m$ as $\eta = (\eta_2, \eta_1, \eta_3)$. The model-based value function $v^m$ and the hindsight value function $v^+$ share the same internal state representation $h$, but the learning of $v^+$ assumes $h$ is fixed (we do not backpropagate through the state-RNN in hindsight). In addition, we force $\hat{\phi}$ to only be learned through $\mathcal{L}_{\mathrm{model}}$, so that $v^m$ uses it as an additional input.[3] Denoting with the bar notation quantities treated as non-differentiable (i.e. where the gradient is stopped) this can be summarized as:

$$v^+(h_t, h_{t+k}; \theta) = \psi_{\theta_1}(\overline{h_t}, \phi_{\theta_2}(\overline{h_{t+k}})), \quad v^m(h_t; \eta) = \psi_{\eta_1}(h_t, \overline{\hat{\phi}_{\eta_2}(h_t)}). \tag{1}$$

The different losses in the HiMo architecture are combined in the following way:

$$\mathcal{L}(\theta, \eta) = \mathcal{L}_v(\eta) + \alpha\mathcal{L}_{v^+}(\theta) + \beta\mathcal{L}_{\mathrm{model}}(\eta). \tag{2}$$

A diagram of the architecture is presented in Fig. 2 (see also details in the appendix). Computing $v^+$ and training $\hat{\phi}$ is done in online fashion by simply delaying the updates by $k$ steps (just like in the computation of an $n$-step return). This architecture can be straightforwardly generalized to cases where we also output a policy $\pi_\eta$ for an actor-critic setup, providing $h$ and $\hat{\phi}$ as inputs to a policy network.[4] For a Q-value based algorithm like Q-learning, we predict a vector of values $q^m$ and $q^+$ instead of $v^m$ and $v^+$. Sec. 6 describes how we use this architecture with both actor-critic and a value-based approaches, including details of the value target $U_t$ and update rules in each scenario.

# 5   Related Work

Recent works used auxiliary predictions successfully in RL as a way to obtain a richer signal for representation learning [10, 20]. However, these additional prediction tasks are hard-coded, so they cannot adapt to the task demand when needed. We see them as a complementary approach to more efficient learning in RL. An exception is the recent work by Veeriah et al. [24], done concurrently to our work, which studies an alternative formulation that employs meta-gradients to discover modelling targets (referred to as "questions").

Buesing et al. [3] have considered using observations in an episode trajectory in hindsight to infer variables in a structural causal model of the dynamics, allowing more efficient reasoning, in a model-based way, about counterfactual actions. However this approach requires learning an accurate generative model of the environment. Other recent work in model-based RL has considered implicit models that directly predict the relevant quantities for planning [18, 13, 16], but these do not exploit the future observations. We see these approaches as complementary to our work. In another recent model-based work by Farahmand [5], the model loss minimizes some form of value consistency: the difference between the value at some next state and the expected value of starting in the previous state. While this makes the model sensitive to the value, it only exploits the future real state through $v$ as a learning signal (just like in bootstrapping). A similar approach around reward consistency is proposed by Gelada et al. [6]. Furthermore, our use of a model output as an input, to be robust to model errors, is inspired by [15].

In supervised learning, the learning using privileged information (LUPI) framework [23] considers ways of leveraging privileged information at train time. Although their approach does not apply directly in RL, some of our approach can be understood in their framework, considering the future trajectory as the privileged information for value prediction. Privileged information coming from full state observation has been leveraged in RL to learn a better critic in asymmetric actor-critic architectures [14, 26]. However this does not use future information and only applies to settings where special side-information (full state) is available at training time.

In Hindsight Credit Assignment (HCA) [9], future information is leveraged in backward models to explain away the noise and improve credit assignment. Although related, our approach differs in how and why we exploit the future information: we aim to facilitate learning a (causal) state representation that would be difficult to learn from returns alone, rather than to understand an action's contribution to an outcome. If HCA can be scaled to larger domains, it would complement our approach nicely. Other RL works exploit hindsight information with a more distant motivation, to counter-factually change the goal as a form of automated curriculum [1].

# 6   Experiments

The illustrative example in Sec. 3.2 demonstrated the positive effect of hindsight modelling in a simple policy evaluation setting. We now explore these benefits in the context of policy optimization in challenging domains: a custom navigation task called Portal Choice, and Atari 2600. To demonstrate the generality and scalability of our approach, we test hindsight value functions in the context of two high-performance RL algorithms: IMPALA [4] and R2D2 [11].

## 6.1   Portal Choice task

The Portal Choice (Fig. 3) is a two-phase navigation task. In phase one, an agent is presented with a contextual choice between two portals, whose positions vary between episodes. The position of the portal determines its destination in phase two, one of two different goal rooms (green and red rooms). Critically, the reward when terminating the episode in the goal room depends on both the color of the goal room in phase two and a visually indicated combinatorial context shown in the first phase. If the context matches the goal room color, then a reward of 2 is given, otherwise the reward is 0 when terminating the episode (see appendix for full details).

An easy suboptimal solution is to select the portal at random and finish the episode in the resulting goal room by reaching the goal pixel, which results in a positive reward of 1 on average. A more difficult strategy is to be selective about which portal to take depending on the context, in order to get the reward of 2 on every episode. A model-free agent has to learn the joint mapping from

contexts and portal positions to rewards. Although the task is not visually complex, the context is combinatorial in nature (the agent needs to count randomly placed pixels) and the joint configuration space of context and portal is fairly large (around 250M). Since the mapping from portal position to rooms does not depend on context, learning the portal-room mapping independently is more efficient.

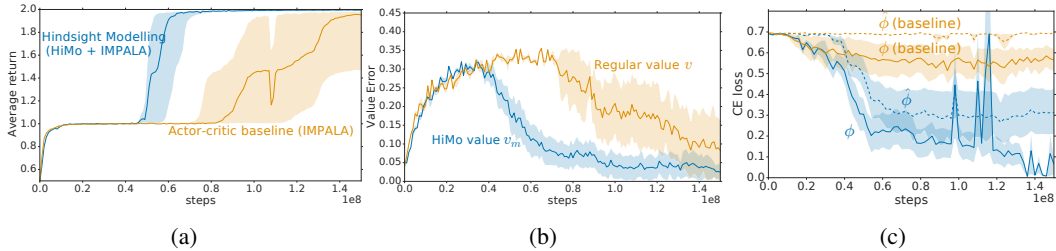

(a)            (b)            (c)

Figure 4: **Portal Choice results.** (a) Median performance as a function of environment steps, out of 4 seeds. (b) shows the value error averaged across states on the same x-axis scale for different value function estimates. (c) shows the cross-entropy loss of a classifier (for analysis) that takes as input $\phi$ (solid line) or $\hat{\phi}$ (dotted line) and predicts the identity of the goal room (red or green) as a binary classification task. The HiMo curves (blue) show that information about the room identity becomes present first in $\phi$ and then gets captured in its model $\hat{\phi}$. For the baseline ($\alpha = \beta = 0$), $\hat{\phi}$ is not trained based on $\phi$ and only classifies the room identity at chance level.

For this domain, we implemented the HiMo architecture within a distributed actor-critic agent named IM-PALA [4]. In this case, the target $U_t$ to train $v^m$ (used as a critic in this context) and $v^+$ is the V-trace target [4], which accounts for off-policy corrections between the behavior policy and the learner policy. The actor ($\pi$) shares the same network as the critic: it also receives $h$ and $\hat{\phi}$ as inputs.

We found that HiMo learned reliably faster to reach the optimal behavior, compared to the vanilla IMPALA baseline that shared the same network capacity (see Fig. 4a). Hindsight Credit Assignment [9] was also tested in this task, but it did not significantly improve the performance beyond IMPALA. With HiMo, the hindsight value $v^+$ rapidly learns to predict whether the portal-context association is rewarding based on seeing the goal room color. To do this, $\phi$ learns to predict the new information from the future which is

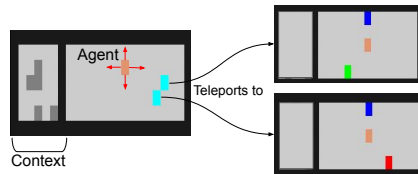

Figure 3: **Portal Choice task.** Left: an observation in the starting room of the Portal Choice task. Two portals (cyan squares) are available to the agent (orange), each of them leading deterministically to a different room, based on their position. Right: The two possible goal rooms are identified by a green and red pixel. The reward upon reaching the goal (blue square) is a function of the room and the initial context.

useful that prediction: the identity of the room (see Fig 4c). The prediction of $\phi$ becomes effectively a model of the mapping from portal to room identity (since the context does not correlate with the room identity). Having access to such a mapping through $\hat{\phi}$ helps the value prediction (Fig 4b), leading to better action selection.

## 6.2 Atari

We tested our approach in Atari 2600 videogames using the Arcade Learning Environment [2]. We added HiMo on top of Recurrent Replay Distributed DQN (R2D2) [11], a DQN-based distributed architecture which previously achieved state-of-the-art scores in Atari games. In this value-based setting, HiMo trains $q^m(\cdot, \cdot; \eta)$ and $q^+(\cdot, \cdot; \theta)$ based on $n$-step return targets: $U_t = g\left(\sum_{m=0}^{n-1} \gamma^m R_{t+m} + \gamma^n g^{-1}\left(q^m(S_{t+n}, A^*; \eta^-)\right)\right)$, where $g$ is an invertible function, $\eta^-$ are the periodically updated target network parameters (as in DQN [12]), and $A^* = \arg\max_a q^m(S_{t+n}, a; \eta)$ (the Double DQN update [22]). Other implementation details are described in the appendix.

We ran HiMo on 57 Atari games for 200k gradient steps (around 1 day of training), with 3 seeds for each game. The evaluation averages the score between 200 episodes across seeds, each lasting a maximum of 30 minutes and starting with a random number (up to 30) of no-op actions. In order to compare scores between different games and aggregate results, we computed normalized scores for each game based on random and human performance, so that 0% corresponds to random

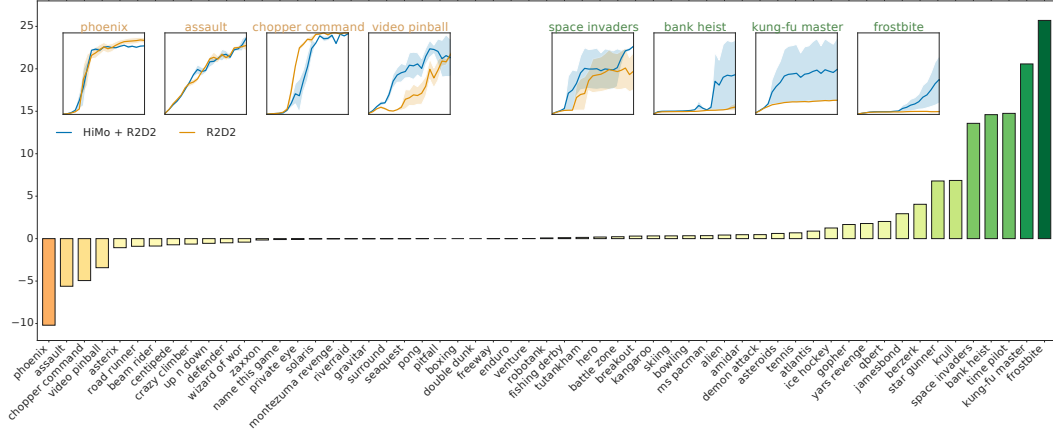

Figure 5: **Comparison of human normalized score in Atari.** Difference in *human normalized score* per game in Atari, HiMo versus the improved R2D2 after 200k learning steps, alongside learning curves for a selection of HiMo worst and top performing games. The high variance of the curves in Atari between seeds can often be explained by the variable timestep at which different seeds jump from one performance plateau to the next.

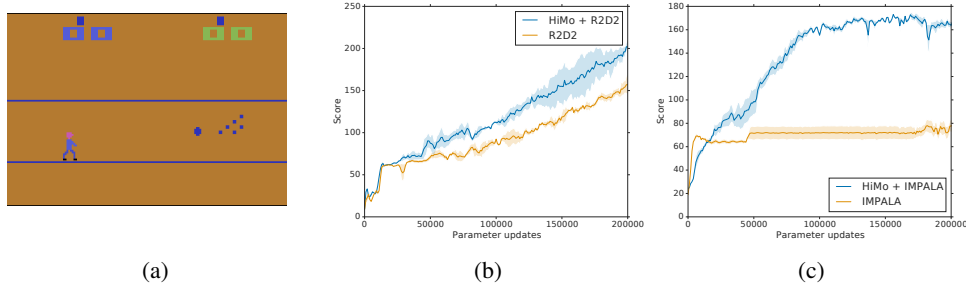

|    (a)    |    (b)    |    (c)    |

Figure 6: **Performance in the Atari bowling game.** In this game, a delayed reward can be predicted by the intermediate event of the ball hitting the pins (a). (b-c) Learning curves for HiMo in two RL settings: a value-based method (R2D2) in (b) and a policy-gradient method (IMPALA) in (c).

performance and 100% corresponds to human. We observed an increase of 132.5% in the median human normalized score compared to the R2D2 baseline with the same network capacity. Aggregate results are reported in Table 1. Fig. 5 details the difference in normalized score between HiMo and our R2D2 baseline for all games individually. We note that the original R2D2 results reported by Kapturowski et al. [11], which used a similar hardware configuration but a different network architecture, were around 750% median human normalized score after a day of training.

We observed that HiMo either offered improved data efficiency or had no overwhelming adverse effects on training performance. In Fig. 5 we show training curves for a selection of representative Atari environments, which seem to indicate that in the worst case scenario, HiMo's training performance reduces to R2D2's.

Bowling is one of the Atari games where rewards are delayed with relevant information being communicated through intermediate observations (the ball hitting the pins), similarly to the baseball example in the introduc-

Table 1: Median and mean human normalized scores across 57 Atari2600 games for HiMo versus the R2D2 baseline after a day of training.

|        | R2D2    | R2D2 + HiMo |
|--------|---------|-------------|
| Median | 832.5%  | **965%**    |
| Mean   | 2818.5% | **2980%**   |

tion. We found that HiMo performed better than the R2D2 baseline in this particular game. We also ran HiMo in the actor-critic setup (IMPALA) described previously, finding similar performance gain with respect to the model-free baseline. These results are presented in Fig. 6.

# 7 Conclusion

High-dimensional observations in the intermediate future often contain task-relevant features that can facilitate the prediction of an RL agent's final return. We introduced HiMo, an RL algorithm that leverages this insight through a two-stage approach. First, by reasoning in hindsight, the algorithm learns to extract relevant features of future observations that would be been most helpful for estimating the final value. Then, a forward model is learned to predict these features and used as input to an improved value function, yielding better policy evaluation at test time. We demonstrated that this approach can help tame complexity in environments with rich dynamics at scale, yielding increased data efficiency and improving the performance of state-of-the-art model-free architectures.

## Broader Impact

This work carries fundamental research in reinforcement learning (RL) using simulated data, with the immediate goal to improve RL techniques. While we are not directly targeting any application domain in this paper, the future impact of this research will be dependent on the context in which such techniques are deployed.

## Acknowledgments and Disclosure of Funding

We thank the anonymous reviewers for their useful feedback.

## Footnotes

[1]See appendix A.1 for a worked out example.

[2]In practice the environment is often partially-observed and the state of the world is not directly accessible. For this case, we can replace the observed state $s$ by a learned function that depends on past observations.

[3]This is motivated by the imagination-augmented agents work by Racanière et al. [15].

[4]In this case, the total loss $\mathcal{L}(\theta, \eta)$ also contains an actor loss to update $\pi_\eta$ and a negative entropy loss.

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
