[Supplementary Material]

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

# A  Appendix

## A.1  Introduction Example

The argument in the introduction follows straightforwardly from a counting argument from the size of the probability tables involved in the discrete case. We now describe a counting argument similar to the discrete probability factorization example in the introduction but that applies more directly to a value function scenario. Consider the following chain:

$$(X, X') \rightarrow (X, Y') \rightarrow Z, \tag{3}$$

where $Z$ is to be interpreted as the expected return. Here the start state $(X, X')$ is sampled randomly but the rest of the chain has deterministic transitions, and $Y'$ is independent of $X$ given $X'$. Let $n$ be the number of possible values of $X$, and $m$ the number of possible values of $X'$, and suppose the number of possible values of $Y'$ is 2. In a tabular setting, learning the start state's value function model-free (i.e. mapping $(X, X')$ directly to $Z$) requires observing returns $Z$ for all $nm$ entries. In contrast, if we estimate the mappings $X' \rightarrow Y'$ and $(X, Y') \rightarrow Z$ separately, it requires $m + 2n$ entries, which is better than $nm$ (for $n, m > 4$). This shows that even in a policy evaluation scenario, the right model can more efficiently learn the value function. The illustrative task in Sec. 3.2 extends this to a function approximation setting but is similar in spirit.

## A.2  Analysis

We restate the proposition from Section 3.3 and prove it here.

**Proposition.** *Suppose that $v_\theta^m$ is sharing the same function $\psi$ as $v^+$ (i.e., $\theta_1 = \eta_1$), and let $\psi$ be linear ($\psi_{\theta_1}(f, \phi) = \begin{pmatrix} \omega_1 \\ \omega_2 \end{pmatrix}^\top \begin{pmatrix} f \\ \phi \end{pmatrix} + b$, where $\theta_1 = (\omega_1, \omega_2)$). Assume a squared loss for the model and the following relation between value losses: $\mathcal{L}(v^+) = C\mathcal{L}(v)$ with $0 < C < 0.5$ (i.e., estimating the value with more information is an easier learning problem). Then, the following holds: $\mathcal{L}_{model}(\eta_2) < \frac{(1-2C)\mathcal{L}(v)}{2\|\omega_2\|^2} \implies \mathcal{L}(v^m) < \mathcal{L}(v)$. (Proof in appendix.)*

*Proof.* We can derive the following relation for fixed values of the parameters:

$$\mathbb{E}[(v^m(s; \eta) - v^+(s, \tau^+; \theta)^2] = \mathbb{E}[|\omega_2^\top (\phi(\tau^+; \theta_2) - \hat{\phi}(s; \eta_2))|^2] \tag{4}$$

$$\leq \mathbb{E}[\|\omega_2\|^2 \|\phi(\tau^+) - \hat{\phi}(s)\|^2] \tag{5}$$

$$= \|\omega_2\|^2 \mathcal{L}_{model}(\eta_2), \tag{6}$$

using the Cauchy-Schwarz inequality. Let $\mathcal{L}$ define the value error for a particular value function $v$: $\mathcal{L}(v) = \mathbb{E}[(v(s) - G)^2]$ and $\mathcal{L}(v^+) = \mathbb{E}[(v^+(s, \tau^+) - G)^2]$. Then we have:

$$\mathcal{L}(v^m) = \mathbb{E}[(v^m(s) - v^+(s, \tau^+) + v^+(s, \tau^+) - G)^2] \tag{7}$$

$$\leq 2(\|\omega_2\|^2 \mathcal{L}_{model}(\eta_2) + \mathcal{L}(v^+)), \tag{8}$$

using the fact that $\mathbb{E}[(X + Y)^2] \leq 2(E[X^2] + E[Y^2])$ for random variables $X$ and $Y$ and equation 6. Using the assumption $\mathcal{L}(v^+) = C\mathcal{L}(v)$ with $0 < C < 0.5$ and equation 8, the result follows:

$$\mathcal{L}_{model}(\eta_2) < \frac{(1 - 2C)\mathcal{L}(v)}{2\|\omega_2\|^2} \implies \mathcal{L}(v^m) < \mathcal{L}(v). \tag{9}$$

$\square$

## A.3  Illustrative Example Details

We provide the precise definition of the illustrative tasks of Sec. 3.2. It consists of a value estimation problem in a 1-step Markov Reward Process (no actions), namely each episode consists of a single transition from initial state $s$ to terminal state $s'$, with a reward $r(s, s')$ on the way. The agent is trained on multiple episodes of each MRP instance (the x-axis in Fig. 1-right). This learning process is repeated independently and averaged for multiple instances of the environment. Each instance of the task is parametrized by a square matrix $W$ and a vector $b$ sampled from a unit normal distribution, as well as randomly initialized MLP (see below), which together determine

the uncontrolled MDP. Initial states $s$ are of dimension $D$ and sampled from a multivariate unit normal distribution ($s_i \sim N(0,1)$ for all state dimension $i$). Given $s = \binom{s_1}{s_2}$, where $s_1$ and $s_2$ are of dimension $D_1$ and $D_2$ ($D = D_1 + D_2$), the next state $s' = \binom{s'_1}{s'_2}$ is determined according to the transition function: $s'_1 = \text{MLP}(s) + \epsilon$ and $s'_2 = \sigma(W s_2 + b)$ where $\sigma$ is the Heaviside function, and MLP is a randomly sampled Multi-Layer Perceptron. $s'_1$ acts as a distractor here, with additive noise $\epsilon \sim N(0,1)$. The reward obtained is $r(s, s') = \sum_i s_1^{(i)} \sum_i s_2'^{(i)} / \sqrt{D}$. The true value in the start state is also $v(s) = r(s, s')$.[5]

Fig. 1-left shows some trajectories for a low-dimensional version of the problem ($D = 4$) and the middle plot displays $s'_2$ by color-coding it according to $\sum_i s_2'^{(i)}$. The results in Fig. 1-right use networks where each subnetwork is a small 1-hidden-layer MLP with 16 hidden units and ReLu activation functions, and $\phi$ has dimension 3. The dimension of the data is $D = 32$ for this experiment, with the dimension of the useful data in the next state $D_2 = 4$.

### A.4   General Architecture Details

To compute $v^+$ and train $\hat{\phi}$ in an online fashion, we process fixed-length unrolls of the state-RNN and compute the hindsight value and corresponding updates at time $t$ if $t + k$ is also within that same unroll. Also, we update $v^+$ at a slower rate (i.e., $\alpha < \beta$) to give enough time for the model $\hat{\phi}$ to adapt to the changing hindsight features $\phi$. In our experiments we found that even a low-dimensional $\phi$ (in the order of $d = 3$) and a relatively short hindsight horizon $k$ (in the order of 5) are sufficient to yield significant performance boosts, whilst keeping the extra model computational costs modest.

For most experiments described in the paper, the model loss $\mathcal{L}_{\text{model}}(\eta_2)$ is the following:

$$\mathcal{L}_{\text{model}}(\eta_2) = \mathbb{E}_{s,\tau^+}[\|\phi_{\theta_2}(\tau^+) - \hat{\phi}_{\eta_2}(s)\|_2^2] \tag{10}$$

where the expectation is taken over the distribution of states and partial trajectories $\tau^+$ resulting from that state. Note that these trajectories $\tau^+$ may be obtained off-policy in some of the experiments (due to the distributed nature of the algorithm, the behavior policy is not fully synchronised with the evaluation policy), and we do not explicitly correct for this effect for simplicity.

### A.5   Portal Choice

**Environment**   The observation is a $7 \times 23$ RGB frame (see Fig. 3). There are 3 possible spawning points for the agent in the center and 42 possible portal positions (half of which lead to the green room, the other half leading to the red room). At the start of an episode, two portals, each leading to a different room, are chosen are random. They are both displayed as cyan pixels. Included in the observation in the first phase is the context, a random permutation in a $5 \times 5$ grid of $N$ pixels, where is uniformly sampled at the start of each episode: $N \sim \mathcal{U}\{1, 10\}$. A fixed map $f : \{1, \ldots, 10\} \to \{0, 1\}$ determines which contexts are rewarding with the green room, the rest being rewarding with the red room. The reward when reaching the goal is determined according to:

$$R = 2(f(N)G + (1 - f(N))(1 - G)), \tag{11}$$

where $G \in \{0, 1\}$ is whether the reached room is green.

Note that the important decision in this task does not require any memory: everything is observed in the portal room to select the portal, yet there is a memory demand when in the reward room. We ran an extra control experiment where we gave $h_{t-k}$ as an additional input to the policy and value for the baseline IMPALA agent and it did not perform better than what is reported in Fig. 4a for the actor-critic baseline.

**Network architecture**   The policy and value network takes in the observation and passes it to a ConvNet encoder (with filter channels [32, 32, 32], kernel shapes [4, 3, 3] applied with strides [2, 1, 1]) before being passed to a ConvLSTM network with 32 channels and 3x3 filters. The output of the ConvLSTM is the internal state $h$. The $\hat{\phi}$ network is a ConvNet with [32, 32, 32, 1] filter channels with kernels of size 3 except for a final 1x1 filter, whose output is flatten and passed to an MLP with

256 hidden units with ReLu activation, before a linear layer with dimension $d = 3$. The $\phi$ network is a similarly configured network with one less convolution layer and 128 hidden units in the MLP. The $\psi_\eta$ network is an MLP with 256 hidden units followed by a linear layer that takes $h$ and $\hat{\phi}$ as input and outputs the policy $\pi^m$ and the value $v^m$. $v^+$ is obtained similarly with a similar MLP that has a single scalar output. We used a future observation window of $k = 5$ steps in this domain and loss weights $\alpha = 0.25$, $\beta = 0.5$. Unroll length was 20, and $\gamma = 0.99$. Optimization was done with the Adam optimizer (learning rate of $5e^{-4}$), with batch size 32. The model-free baseline is obtained by using the same code and network architecture, and setting the modeling loss and hindsight value loss to 0 ($\alpha = \beta = 0$).

For the portal task, we found it better to employ a cross-entropy loss for the model loss:

$$\mathcal{L}_{\text{model}}(\eta_2) = \mathbb{E}_{s,\tau^+}[H(p(\phi_{\theta_2}(\tau^+)), \hat{p}(\hat{\phi}_{\eta_2}(s)))] \tag{12}$$

where $p^{(i)} \propto e^{\phi_{\theta_2}(\tau^+)^{(i)}}$ and $\hat{p}^{(i)} \propto e^{\hat{\phi}_{\eta_2}(s)^{(i)}}$ are the softmax distributions when interpreting $\phi$ and $\hat{\phi}$ as the vector of logits.

## A.6  Atari

Hyper-parameters and infrastructure are the same as reported in [11], with deviations as listed in table 2. For our value target, we also average different $n$-step returns with exponential averaging as in $Q(\lambda)$ (with the return being truncated at the end of unrolls). The $Q$ network is composed of a convolution network (cf. Vision ConvNet in table) which is followed by an LSTM with 512 hidden units. What we refer to in the main text as the internal state $h$ is the output of the LSTM. The $\phi$ and $\hat{\phi}$ networks are MLPs with a single hidden layer of 256 units and ReLu activation function, followed by a linear which outputs a vector of dimension $d$. The $\psi_{\theta_1}$ function concatenates $h$ and $\phi$ as inputs to an MLP with 256 hidden units with ReLu activation function, followed by a linear which outputs $q^+$ (a vector of dimension 18, the size of the Atari action set). $q^m$ is obtained by passing $h$ and $\hat{\phi}$ to a dueling network as described by [11].

Other HiMo parameters are described in table 3. The R2D2 baseline with the same capacity is obtained by running the same architecture with $\alpha = \beta = 0$.

Table 2: Hyper-parameter values used for our R2D2 implementation.

| | |
|---|---|
| Number of actors | 320 |
| Sequence length | 80 (+ prefix of l = 20 in burn-in experiments) |
| Learning rate | $2e^{-4}$ |
| Adam optimizer $\beta_1$ | 0.9 |
| Adam optimizer $\beta_2$ | 0.999 |
| $\lambda$ | 0.7 |
| Target update interval | 400 |
| Value function rescaling | $g(x) = \text{sign}(x)\left(\sqrt{\|x\| + 1} - 1\right) + \epsilon x,\ \epsilon = 10^{-3}$ |
| Frame pre-processing | None (full res. including no frame stacking) |
| Vision ConvNet filters sizes | [7, 5, 5, 3] |
| Vision ConvNet filters strides | [4, 2, 2, 1] |
| Vision ConvNet filters channels | [32, 64, 128, 128] |

Table 3: Hindsight modelling parameters for Atari

| | |
|---|---|
| $\alpha$ | 0.01 |
| $\beta$ | 1.0 |
| $k$ | 5 |
| $d$ | 3 |

Table 4: Score details per game in Atari. Results for 200k updates, as summarized in Table 1.

| Level name | HiMo (+R2D2) | R2D2 (our run) | Human | Random |
|---|---|---|---|---|
| alien | 95,450.38 | 92,532.38 | 7,127.70 | 227.80 |
| amidar | 23,547.32 | 22,754.29 | 1,719.50 | 5.80 |
| assault | 83,560.22 | 86,476.19 | 742.00 | 222.40 |
| asterix | 988,988.10 | 997,833.33 | 8,503.30 | 210.00 |
| asteroids | 170,831.05 | 142,393.95 | 47,388.70 | 719.10 |
| atlantis | 1,420,511.90 | 1,406,197.62 | 29,028.10 | 12,850.00 |
| bank heist | 13,659.95 | 2,869.57 | 753.10 | 14.20 |
| battle zone | 641,233.33 | 633,438.10 | 37,187.50 | 2,360.00 |
| beam rider | 160,192.84 | 174,640.44 | 16,926.50 | 363.90 |
| berzerk | 19,161.86 | 9,026.14 | 2,630.40 | 123.70 |
| bowling | 205.04 | 158.33 | 160.70 | 23.10 |
| boxing | 100.00 | 100.00 | 12.10 | 0.10 |
| breakout | 777.57 | 768.95 | 30.50 | 1.70 |
| centipede | 523,788.61 | 530,994.40 | 12,017.00 | 2,090.90 |
| chopper command | 967,389.05 | 999,875.24 | 7,387.80 | 811.00 |
| crazy climber | 257,869.05 | 273,993.81 | 35,829.40 | 10,780.50 |
| defender | 478,032.38 | 485,789.05 | 18,688.90 | 2,874.50 |
| demon attack | 143,439.98 | 142,584.74 | 1,971.00 | 152.10 |
| double dunk | 24.00 | 24.00 | -16.40 | -18.60 |
| enduro | 2,367.29 | 2,365.93 | 860.50 | 0.00 |
| fishing derby | 73.45 | 67.36 | -38.70 | -91.70 |
| freeway | 32.97 | 32.96 | 29.60 | 0.00 |
| frostbite | 119,402.71 | 9,669.95 | 4,334.70 | 65.20 |
| gopher | 114,422.95 | 110,841.14 | 2,412.50 | 257.60 |
| gravitar | 6,759.05 | 6,826.19 | 3,351.40 | 173.00 |
| hero | 27,729.79 | 22,108.21 | 30,826.40 | 1,027.00 |
| ice hockey | 45.19 | 30.04 | 0.90 | -11.20 |
| jamesbond | 13,305.00 | 12,501.79 | 302.80 | 29.00 |
| kangaroo | 14,430.48 | 13,475.11 | 3,035.00 | 52.00 |
| krull | 104,218.76 | 96,904.48 | 2,665.50 | 1,598.00 |
| kung-fu master | 673,359.79 | 210,872.38 | 22,736.30 | 258.50 |
| montezuma revenge | 133.33 | 266.67 | 4,753.30 | 0.00 |
| ms pacman | 29,259.33 | 26,924.95 | 6,951.60 | 307.30 |
| name this game | 38,484.52 | 39,106.19 | 8,049.00 | 2,292.30 |
| phoenix | 733,645.10 | 799,754.44 | 7,242.60 | 761.40 |
| pitfall | 0.00 | 0.00 | 6,463.70 | -229.40 |
| pong | 20.99 | 21.00 | 14.60 | -20.70 |
| private eye | 10,138.16 | 16,690.32 | 69,571.30 | 24.90 |
| qbert | 107,484.88 | 80,638.69 | 13,455.00 | 163.90 |
| riverraid | 36,276.24 | 36,627.05 | 17,118.00 | 1,338.50 |
| road runner | 545,206.67 | 552,289.52 | 7,845.00 | 11.50 |
| robotank | 81.78 | 81.01 | 11.90 | 2.20 |
| seaquest | 999,403.80 | 999,934.59 | 42,054.70 | 68.40 |
| skiing | -25,551.48 | -29,694.19 | -4,336.90 | -17,098.10 |
| solaris | 3,902.95 | 4,342.38 | 12,326.70 | 1,236.30 |
| space invaders | 56,694.81 | 36,046.48 | 1,668.70 | 148.00 |
| star gunner | 357,928.57 | 292,842.38 | 10,250.00 | 664.00 |
| surround | 9.68 | 9.93 | 6.50 | -10.00 |
| tennis | 15.79 | 5.19 | -8.30 | -23.80 |
| time pilot | 264,966.19 | 240,458.10 | 5,229.20 | 3,568.00 |
| tutankham | 341.03 | 318.57 | 167.60 | 11.40 |
| up n down | 469,491.38 | 475,690.81 | 11,693.20 | 533.40 |
| venture | 1,981.43 | 1,966.19 | 1,187.50 | 0.00 |
| video pinball | 660,922.63 | 721,413.80 | 17,667.90 | 0.00 |
| wizard of wor | 97,959.05 | 99,697.14 | 4,756.50 | 563.50 |
| yars revenge | 494,489.72 | 402,853.37 | 54,576.90 | 3,092.90 |
| zaxxon | 87,735.24 | 89,230.95 | 9,173.30 | 32.50 |