[Reviews · NeurIPS 2020]

Review 1

Summary and Contributions: This paper presents a value function architecture, hindsight model (HiMo), that utilizes future states of trajectory to make the learning problem easier at training time. It additionally learns a predictive model of (the embedding of) future states, which is used at test time when future states are not available. The proposed method is relatively novel, and the results show that the method is quite effective at improving the policy learning over R2D2 on the Atari benchmark, as well as two simpler domains.

Strengths: The paper presents an interesting novel idea and the results are strong. The first illustrative example nicely shows how value prediction is easier with the additional knowledge of future states, and section 3.3 provides a nice intuitive theorem on how value prediction is easier given extra information in a linear setting. Next, the “portal choice” experiment shows that in an environment with particularly combinatorial structure, HiMo takes advantage of the observed future states for value prediction. Finally, the results on Atari show that HiMo improves average performance over the 57 games quite significantly over R2D2.

Weaknesses: One issue with the method is that it induces a significant amount of complexity, with many new components and losses. However, the effectiveness on the range of Atari tasks is reassuring that the models are not too finnicky to train. Another issue I see is that future states depend on future actions; thus by using HiMo even off-policy algorithms are transformed to be somewhat on-policy, but this is not explicitly handled or discussed. In fact, given this mismatch, it is actually quite surprising that this works out of the box without an explicit correction, so a discussion of this issue would be appreciated.

Correctness: No issues with the methodology.

Clarity: The writing is well-written, organized, clear, and easy to read.

Relation to Prior Work: The relation to prior work is quite discussed clearly. There is a strong connection to Imagination-Augmented Agents for Deep Reinforcement Learning (Weber 2017), which is not referenced, but 3 other related papers from the same authors are referenced. Also, while not necessarily fair to ask for as a baseline, a comparison against this line of work and/or using the model-based value function (presented in Figure 1) rather than a bare RL algorithm like R2D2 or IMPALA would help contextualize the work better.

Reproducibility: Yes

Additional Feedback: Minor comments: - Figure 2 is hard to read because of the notation; please explain it in the caption so the reader doesn’t have to flip back and forth between the figure and the method section. == After rebuttal == I am happy with the rebuttal response and I think this is a valuable contribution.


Review 2

Summary and Contributions: This paper describes a method for learning value functions assuming access to privileged information of future states in a trajectory. Because such information is obviously not available at test time, an approximation to this future prediction must be learned, which takes the form of a low-dimensional "summary" of future states with reward-relevant information. At an architectural level, this could be viewed as a particular parametrization of a value function; however, its usefulness comes from supervising the hindsight model with a signal different from the usual bootstrapped target value. This allows the value function to use supervision from future states in a way that is reminiscent of model-based methods without directly predicting the future, which many have observed to be challenging in high-dimensional state spaces.

Strengths: It has been surprisingly difficult to scale model-based RL methods to some of the domains that are now regularly tackled with model-free methods (like the Atari benchmark used in this paper), but when model-based methods can be made to work they are often observed be somewhat more sample-efficient. This makes the general question studied in the paper (of how to best leverage model-like predictions without suffering from overwhelming reconstruction errors in high dimensions) timely and important. The empirical evaluation includes both small-scale experiments to build intuition and a large-scale study, showing that the proposed method substantially improves over R2D2 for a small number of domains and is mostly a wash on the rest, so there is little downside given that the additional architectural components and supervision are not too costly.

Weaknesses: This paper fairly convincingly argues that supervision beyond the reward function can be leveraged effectively by an RL algorithm without resorting to model-based predictions in observation space. However, this point is not necessarily a new one, so the most pressing question is: how does HiMo compare to other methods that are motivated by the exact same intuition? Most of the relevant points of comparison are included in the related work section (auxiliary tasks, value prediction network, and Mu-Zero are mentioned; successor features and descendants also seem relevant but are missing), and these are argued to be either (a) complementary or (b) "not exploit[ing] the future observations". I agree that they are implementationally different, but given that they have the same goal and operate under rather similar high-level motivations, it would be useful to know how these approaches compare. In general, I think a comparison to other pseudo-model-based methods like Mu-Zero would carry much more information than a comparison to R2D2, so given the scale of the experimental study I would rather see evaluation on fewer domains (eg, only half of the Atari-57 suite) but with more relevant baselines so that readers could better understand how algorithms in this space relate.

Correctness: The proof of the proposition provided in Appendix A2 appears correct to me.

Clarity: The paper is clearly written.

Relation to Prior Work: Relevant prior work is discussed, but an empirical comparison to closely-related algorithms would make the empirical evaluation much stronger (see "Weaknesses" above).

Reproducibility: Yes

Additional Feedback: I lean toward a weak accept given the clear presentation of a timely idea and convincing results compared to a fully model-free method. I think the paper would be made stronger by comparing to other pseudo-model-based ideas that are motivated by the same intuition.


Review 3

Summary and Contributions: This paper introduces an approach in between model-based and model-free RL by using hindsight to reason about what parts about the future could have been useful in the past. Their approach is different from bootstrapping in that it gives more information about the future through some features rather than just a return value. Their method is tested in 57 Atari environments and shows improvements in performance and faster convergence to optimality.

Strengths: - Hindsight modeling achieves a balance between model-based and model-free learning, and achieves significantly better performance when added to 2 well-performing RL algorithms. The results of this work show a fairly concrete improvement to two vanilla RL algorithms we currently use, and perhaps using the hindsight idea they’ve proposed will show improvements in other algorithms as well. - Motivation is pretty clear, and explains why hindsight modeling helps and why it’s better than bootstrapping. Hence I am convinced this is a promising and exciting idea. - Results show pretty significant performance improvements over SOTA. - Seems to improve on prior work on modeling w.r.t future states (Hindsight Credit Assignment experiments were run on very toy envs, and here it is atari) - Toy environment is fairly convincing for intuition.

Weaknesses: - "In addition, we force \hat{\phi} to only be learned through L_model, so that vm uses it as an additional input" (189) seems unclear / unmotivated. - Atari normalized score looks standard, and not much insight gained into the strengths of the hindsight value functions. - What is Y_t in figure 2, maybe this figure could be better represented?

Correctness: Overall, looks correct.

Clarity: The presentation and structure are fairly clear, starting off with what hindsight is and why it’s useful in this setting (other than related work section being at the end) However, the notations are a little hard to follow.

Relation to Prior Work: Moving the related work section up towards the beginning might be helpful in better understanding hindsight and other related work that’s been done in this area, also makes more sense when comparing to Hindsight Credit Assignment in experiments

Reproducibility: Yes

Additional Feedback: I have read the author response and the reviews from other authors. I'm satisfied with the response and stick to my original score of Accept.


Review 4

Summary and Contributions: Update: After reviewing the author feedback I still feel as though this paper is lacking in real insights, and the theoretical contribution is quite limited. The paper provides some good intuitions and discussion for why the approach may help, but the core contribution is really mainly a slight empirical performance boost in Atari. =========== The authors introduce a method for learning value function features by predicting features of the future trajectory. This can be used either for value-based methods or as a baseline for policy gradient methods. In their practical implementation, they use an RNN based approach. The features of the future trajectory are a function of the future hidden state, h_{t+k}. A second function approximator tries to predict these features using only h_t. They implemented an IMPALA-based actor-critic algorithm and an R2D2-based value-based algorithm. Their actor-critic algorithm learned faster than vanilla IMPALA on a custom task designed to highlight the benefits of hindsight modeling. The value-based algorithm showed modest gains on Atari.

Strengths: The approach is reasonable. There has been a great deal of interest in representation learning for value functions, so the goals of the paper are relevant to the community. The specific approach has not been tried before to my knowledge.

Weaknesses: The paper was pretty light on theory. While they provide some intuition for their approach, and a fairly modest “proposition,” it’s not clear that this paper significantly advances the field’s understanding of value-function estimation. The experiments did not show any kind of massive benefit. The effect size on the Atari environments, for example, was a small improvement in the human-normalized performance from 2818.5% to 2980%. Such a small effect size calls for a statistical hypothesis test and uncertainty quantification, otherwise it is hard as a reviewer to know whether the result is even repeatable. On the other hand, it’s not clear that small effects like this are of great interest to the community to begin with. Many researchers I talk to in the field specifically complain about incremental results on Atari. In the only other results in the paper, the proposed algorithm does work faster, but the baseline still converged to the optimal performance.

Correctness: The method is certainly valid within the RL paradigm. As mentioned before, the empirical methodology is lacking any quantification of the uncertainty. While the role of statistical hypothesis testing in RL is hotly contested, I think it’s pretty widely agreed that it is necessary when the effect size is small/debatable. In this case, I do not feel as though I can say with certainty that the author’s approach outperformed the baseline.

Clarity: The paper is reasonably clear and well-written. In American English, “modeling” is typically spelled with a single “L.” I’m not sure what NeurIPS’s opinion on this is, however.

Relation to Prior Work: The authors did a reasonable job discussing prior work. One paper that may be worth mentioning is “Value Prediction Network” by Oh et al., which incorporates some similar ideas.

Reproducibility: Yes

Additional Feedback: I would suggest trying to devise experiments that do not merely show learning curves, but somehow demonstrate why the features learned by the algorithm are interesting. Unless you can get a really big effect, many in the community do not feel that the incremental performance gains aren’t really of great interest.

[Author Response · NeurIPS 2020]

We thank all reviewers for their time and useful feedback.

**General point about Figure 2:** We apologize for a typo in the figure. $Y_t$ should be $U_t$ as defined in Section 2 (the
value target). There was a late notation change that wasn't reflected in the figure during edits. Thanks to reviewer R1-3
for comments about the figure, we'll improve clarity as well.

**R1:**
• **About off-policyness:** This is a great point. In our experiments, the model targets the expected $\phi$ under the
behavior policy (which is different but close to $\pi$), so this model target could be different from the expected $\phi$
under $\pi$. However, even without any correction, this is a valid model of the future which we learn to interpret
for better value predictions (Note: the hindsight and normal value weights are not shared in this case, $\theta_1 \neq \eta_1$).
Different correction schemes are possible, but we wanted to keep the approach simple; we'll add a note in
the paper. One related note: R2D2 computes $n$-step returns without any correction for off-policyness, but
IMPALA corrects the value targets with VTrace.

• **"(Weber 2017),"** We apologize, that's an omission and we meant to cite it since it inspired some of the design
choices (using the model as feature).

**R2:**
• **about baselines:** The reason we selected model-free baselines was that 1) our method in a way sits between
classical model-free and model-based methods and 2) model-free methods have been dominating empirically in
many domains. So we wanted to see whether our approach could contribute beyond the best performing method.
Given the recent results of MuZero in Atari, that would also be a good candidate but 1) we cannot directly
compare fairly against the published results since any difference in the setup/network architecture would
not provide a well-controlled experiment and 2) MuZero is considerably more expensive to run than R2D2
compute-wise to reproduce in a comparable setting. Nevertheless, we have started a broader experimental
investigation by providing a comparison to a model-based approach in Fig 1 and preliminary results of HCA
in Portal choice (where it did not perform well). And we plan to study more comparisons in the future.

**R3:**
• **"(189) seems unclear / unmotivated."** This is motivated by the Imaginative Agents line of work (Weber et al.
2017) of using the model output as an input feature to an agent. We'll expand to make this clearer.

• **"Moving the related work section up"** That's a good suggestion, we'll move it.

**R4:**
• **"Unless you can get a really big effect, ..."** Many major results in our field were achieved by combining
'incremental' performance gains together. Our approach, which all reviewers agreed is novel, is evaluated in
two smaller scale domains where the performance is clearly superior to the baseline (Fig 1R, Fig 4a) and where
it's feasible to provide a detailed empirical analysis. The goal of the Atari experiment was to demonstrate
that the same setup could work at scale in combination with SOTA RL agents. Our results show that the
proposed approach combines easily with these agents and it significantly improves their performance in many
games. Note that R2D2 already gets the maximum score in a number of games, so there is a ceiling to possible
improvements. Nonetheless, the aggregate median metric shows an overall improvement (832.5% to 965%)
and the detailed scores per game show that the performance improved significantly in a number of games.

• **"demonstrate why the features learned by the algorithm are interesting"** An analysis of the features
learned by HiMo in the Portal Choice domain is already provided in Figure 4-c. It shows that the features
capture the room color identity in practice, which is exactly the right information to model in hindsight.

• **"quantification of the uncertainty"** All individual domains
   were evaluated using multiple seeds, and the corresponding
   plots indicate uncertainty intervals. We have additionally up-
   dated our main Atari result, which aggregate results across
   games, to also quantify the uncertainty. This was done by ap-
   plying bootstrap sampling to the evaluation episodes across the
   3 seeds (repeated 5000 times), resulting in 95% CI intervals
   of [941.05%-1028.67%] for HiMo's median normalized score,
   and [832.5%-838.38%] for the R2D2 baseline's median score.
   A 2-sided K-S test allows us to reject the hypothesis that these
   were drawn from the same underlying distribution ($p \leq 1e^{-10}$)

*Empirical median distributions in Atari obtained from a bootstrap procedure.*

• **"algorithm does work faster, but the baseline still converged to the optimal performance."** R4 acknowl-
edges that our algorithm learns faster, but this is exactly our claim. Vanilla model-free RL methods will
converge to the optimal performance eventually (modulo some assumptions) but they may be slow getting
there. Our approach attempts to leverage the same trajectory data in a more effective way to improve the
learning process. This is what we demonstrate in the experiments.

• **"Value Prediction Network"** Please note this paper is already cited, cf. [13].

[Meta-Review · NeurIPS 2020]

Learning value functions is a central theme in reinforcement learning. It is a hard problem because of the non-stationary nature of bootstrapping. This paper proposes a fresh approach for improving the learning of value functions by conditioning them on some information of the future states at training time (hindsight). Conditioning on the right future data should provide more certainty about the future return. All the reviewers liked the premise of the paper, clear motivation, and thorough experiments. Reviewer's raised some good technical questions about reliance on trajectory even for off-policy methods, etc. The authors' provided a thoughtful rebuttal addressing these concerns. The paper was discussed in the post-rebuttal phase and everyone agreed that the paper provides interesting insights to be shared with the community. Please refer to reviewers' final comments and incorporate their responses in the camera-ready version.